# GPCRs in Intracellular Compartments: New Targets for Drug Discovery

**DOI:** 10.3390/biom12101343

**Published:** 2022-09-22

**Authors:** Irene Fasciani, Marco Carli, Francesco Petragnano, Francesco Colaianni, Gabriella Aloisi, Roberto Maggio, Marco Scarselli, Mario Rossi

**Affiliations:** 1Department of Biotechnological and Applied Clinical Sciences, University of L’Aquila, 67100 L’Aquila, Italy; 2Department of Translational Research on New Technologies in Medicine and Surgery, University of Pisa, 56126 Pisa, Italy

**Keywords:** G protein-coupled receptors, nuclear membrane, mitochondria

## Abstract

The architecture of eukaryotic cells is defined by extensive membrane-delimited compartments, which entails separate metabolic processes that would otherwise interfere with each other, leading to functional differences between cells. G protein-coupled receptors (GPCRs) are the largest class of cell surface receptors, and their signal transduction is traditionally viewed as a chain of events initiated from the plasma membrane. Furthermore, their intracellular trafficking, internalization, and recycling were considered only to regulate receptor desensitization and cell surface expression. On the contrary, accumulating data strongly suggest that GPCRs also signal from intracellular compartments. GPCRs localize in the membranes of endosomes, nucleus, Golgi and endoplasmic reticulum apparatuses, mitochondria, and cell division compartments. Importantly, from these sites they have shown to orchestrate multiple signals that regulate different cell pathways. In this review, we summarize the current knowledge of this fascinating phenomenon, explaining how GPCRs reach the intracellular sites, are stimulated by the endogenous ligands, and their potential physiological/pathophysiological roles. Finally, we illustrate several mechanisms involved in the modulation of the compartmentalized GPCR signaling by drugs and endogenous ligands. Understanding how GPCR signaling compartmentalization is regulated will provide a unique opportunity to develop novel pharmaceutical approaches to target GPCRs and potentially lead the way towards new therapeutic approaches.

## 1. Introduction

The best current estimate claims that cells appeared on Earth ~3.0–3.3 billion years ago [1]. The process of cellularization is still highly debated, with specialists suggesting that cells were originated by the evolution of the cytoplasm inside a primordial lipid vesicle, while others argue the possibility that the cytoplasm was instead developed outside the surface of early vesicles, and thus, cells were simply the result of an invagination process of this proto-cytosol into those vesicles that had an early functional cytoskeleton [2]. The acquisition of the cell membrane was the first crucial step in biological evolution that defined the boundaries of a cell and allowed controlled communications between intracellular and extracellular environments to produce those relative stable equilibriums crucial for life to evolve. In fact, besides its function as structural support and defense, the cell membrane regulates exchanges of molecules from and to the external environment, responding to environmental physical and chemical changes, and thus plays a crucial role in the interactions with other cells and/or the extracellular environments; moreover, the cell membranes are involved in energy formation and other chemical reactions. Notably, even though some prokaryotes also contain intracellular structures that can be seen as primitive organelles, prokaryotes and archaea have maintained the mono-compartmental structure of their ancestor cells [3]. In fact, it was the evolution from this simple membrane organization into complex compartmentalized eukaryotic cell structures that profoundly marked the second fundamental step in the history of biology: the ability to isolate different types of enzymatic and chemical reactions into intracellular membrane compartments that best favored the different types of cell processes. Delimited and chemically adjusted intracellular microenvironments in fact increased reaction efficiency and limited the interference from other cell processes, which allowed the evolutionary process to proceed towards increasingly complex organisms.

## 2. Evolution Promoted Biological Complexity by Compartmentalization of Metabolic Processes

A simple eukaryote cell shows more complexity than the most complex prokaryote. The main difference lies in the eukaryote cell ability to compartmentalize the metabolic reactions necessary for survival. Eukaryotes have developed subcellular structures separated from each other to confine specific reactions to suitable cell compartments and allow different reactions to take place simultaneously inside the same cell [4]. Intracellular functional specialization and development of pluricellular organisms with increasingly complex and specialized cells were the key features to maximize defense and take advantage of environmental resources more effectively [5]. 

Given the remarkable complexity jump forward between prokaryotes and eukaryotes and the absence of organisms with an intermediate complexity, intermediate cell compartmentalization processes are difficult to identify. One of the main hypotheses, the endosymbiotic theory, proposes that the intracellular compartments (organelles) derived from different organism interactions to perform distinctive reactions such as oxidative respiration and detoxification [6]. This high degree of complexity requires a tight regulation especially for operating and maintaining the functions of these organelles. Notably, each organelle has different protein sets that allow specific reactions to take place without interfering with the reactions in other organelles or the cytosol. Moreover, considering that the genetic information is kept in the nucleus and the protein synthesis happens in the cytoplasm (except for mitochondrial proteins) the targeting of proteins to different cell locations is considered one of the most crucial steps in eukaryotic evolution [7]. However, protein sorting to specific subcellular compartments raises important points. For instance, specific protein sorting to a compartment implies that the targeting signals attached to the protein sequence and the entire sorting process were subjected to evolutionary pressure, including those soluble receptors/interacting proteins that recognize these signals and deliver the proteins to their specific intracellular compartments. This process of signal appearance and recognition happened over a long period of time, protein by protein, till whole pathways were completely compartmentalized. Moreover, these evolutionary processes were not efficient at the beginning and the protein targeting signals might have led to the delivery of proteins into unwanted cell compartments where they were not useful. However, this inefficient protein sorting would also be the cause of the gain of new functions by different organelles [8] allowing the evolution process to select the most advantageous combinations of enzymes and compartments. These interactions are probably the foundation of the extraordinary ability of eukaryotes to take advantage and maximize the environmental resources [9,10]. The ability to sustain functionally separate cell compartments in general requires a precise control of the molecular machinery needed for assembly, maintenance, and inheritance of diverse trafficking processes. The more regulated and articulated those processes are, the more complex the organism becomes. This increased compartmentalization accounts even for the de novo evolution process that allows for new functions, new adaptations, and novel specialization processes to take place and thus for the formation of increasingly highly complex organisms.

## 3. G protein-Coupled Receptors (GPCRs) and Their Role in Modulating Different Types of Stimuli

GPCRs are the largest family of membrane receptors, with nearly 800 genes coding for these proteins. They are involved in many physiological processes such as sensing light, taste and smell, neurotransmission, metabolism, endocrine and exocrine secretion, cell growth, and migration. In a simplistic way, GPCRs are composed of seven transmembrane domains, three extracellular and three intracellular loops, an extracellular N-terminus, and an intracellular C-terminus [11]. The extracellular loop and the extracellular portion of the seven transmembrane domains are deputed to agonist recognition, while the rest of the transmembrane core and the intracellular loops convey the receptor activation to intracellular transducers.

The evolutionary success of GPCRs is thought to depend upon the interplay between the distinctive properties of their extracellular domains to be readily adaptable for new sensory functions, as shown by the ability to be activated by such different types of stimuli as photons, odorants, neurotransmitters, and hormones, and the conservation of a transmembrane core and intracellular signal transduction mechanisms [12,13]. In fact, most of the vertebrate physiology is based on the signal transduction of GPCRs. G proteins are the first and more characterized partners of GPCRs. They are trimeric proteins composed of α, β, and γ subunits and are activated by the nucleotide GTP. The α and β/γ subunits dissociate upon GPCR activation and are regarded as two independent functional units that regulate different downstream effectors [14]. To date, eighteen α, five β, and twelve γ G protein subunits have been discovered. Based on their similarity and functional activities, the α subunits are further subdivided into four different families: (i) Gαs, that stimulates adenylyl cyclase; (ii) Gαi that inhibits the activity of adenylyl cyclase; (iii) Gαq, whose activation stimulates the phospholipase enzyme; and (iv) Gα12 that activates the small GTPase Rho. Instead, Gβ/γ subunits form a tightly bound dimeric complex that plays a critical role in regulating the catalytic activity of the G protein α subunits, and in modulating the activity of several enzymes and ion channels [15]. In contrast to G protein α subunits, the βγ do not have a catalytic activity on their own but modulate signaling through protein–protein interactions. The first effector found to be activated by Gβγ was the inwardly rectifying K^+^ channel (GIRK) in atrial myocytes after muscarinic M2 receptor activation [16,17]. At the beginning, the idea that βγ dimers alone could be the primary mediators of signal activation was controversial, but today, the list of Gβγ interacting proteins comprises a substantial number of targets, such as enzymes and channels [15]. It is worth mentioning that G proteins can also be activated in a GPCR-independent way [18]. Receptor-independent activators of G protein signaling (AGS) play surprising roles in signal processing and have opened new areas of research related to the role of G proteins in signal transduction [19].

The functional complexity of GPCRs could not be attributed to only the exclusive activation of G proteins. Evidence has gradually emerged that GPCRs can signal through many other proteins such as β-arrestins and small G proteins, among others [20]. Notably, by means of a membrane yeast two-hybrid system, it has been recently shown that GPCRs can form interactomes connecting more than 686 proteins that regulate diverse cellular functions [21].

## 4. GPCRs Are Present in Different Cellular Compartments

GPCRs were originally thought to exclusively localize to the plasma membrane and to mediate cellular signaling of stimuli coming from outside the cell. Even though early evidence suggested a subcellular localization and function of some GPCRs, the interest in these “unusual” locations was scarce as it was assumed that activation by ligands was restricted to the plasma membrane. Progress in this area finally demonstrated that GPCR-mediated signaling occurs not only from the plasma membrane but also from intracellular compartments such as endosomes, Golgi membranes [22], mitochondria [23], cell division compartments (centrosomes, spindle midzone, and midbodies) [24], and nuclear membrane [25] (Figure 1 and Table 1).

The functional importance of GPCR subcellular localization was originally shown for the rhodopsin receptor [52,53] that primarily activates the G proteins from the intracellular disk membranes of the rod cell outer segment. In particular, the disk membranes originated from basal evaginations of the plasma membrane of the rod cell outer segment that were retained intracellularly. The subcellular segregation is crucial for the optimal response of rhodopsin to light, as the disk membrane contains six times less cholesterol than the plasma membrane. Rhodopsin is also the major protein of the plasma membrane of rod cells, but the high membrane cholesterol content inhibits rhodopsin participation in the visual transduction cascade at this site [52].

Further evidence suggesting that GPCR signaling also occurred within an internal membrane compartment emerged in studies of β2 adrenergic receptor-mediated activation of the mitogen-activated protein kinases (MAPKs) Erk1/2, which was inhibited by dominant-negative versions of dynamin and β-arrestin-1 that by preventing receptor internalization showed that β2 adrenergic compartmentalized signaling was responsible for ERK activation [54]. Furthermore, overexpression of a mutant β-arrestin-1 that binds c-SRC (a non-receptor tyrosine kinase that regulates the RAS-MAPK/Erk signaling) without promoting β2 adrenergic receptor internalization also shows a reduced phosphorylation of ERK mediated by β2 adrenergic receptor activation, supporting the concept that this receptor signaling occurred within an internal compartment of the cell [55]. Several GPCRs have been known to colocalize on endocytic vesicles together with β-arrestins and Erk1/2, strongly suggesting that endosomes are specific compartments for GPCR–β-arrestin-mediated signaling [22]. Recent evidence suggests that signaling from endosomes may participate in the pathogenesis of cardiac diseases. After prolonged isoproterenol stimulation, β_1_ adrenergic receptors decrease on the cell surface by endocytosis, but they are still active and as shown by Morisco et. al., they mediate cardiac hypertrophy [56]. In fact, the inhibition of the β_1_ adrenergic receptor internalization process by concanavalin A blocks isoproterenol-induced cardiac hypertrophy, strongly suggesting that the cardiac hypertrophy was caused by the ability of these receptors to signal from endosome compartments.

The Golgi apparatus is another compartment where ligand-dependent GPCR activation has been detected. For instance, the β1 adrenergic receptor is known to activate the Gαs protein canonically in the plasma membrane, but also in the Golgi apparatus where it is sorted to form a pre-existing receptor pool [57]. Physiologically, catecholamines are charged and need transporters to cross the plasma membrane such as organic cation transporter 3 (OCT3) to reach for example the Golgi β1 adrenergic receptor [57]. This internal membrane pool of receptors contributes significantly to the overall production of cellular cAMP elicited by β_1_ adrenergic agonists.

Among other receptors activated in the Golgi apparatus, there is the TSH receptor. In this case, the receptor is internalized together with its agonist and when it reaches the Golgi network it activates a local pool of Gαs protein. This critical receptor localization near the nucleus seems required for efficient CREB phosphorylation and gene transcription [58].

Another internal compartment where GPCRs are localized and function is the mitochondrion. Early evidence of this localization came from the cannabinoid CB1 receptor that was shown to be sorted in the outer mitochondrial membrane of skeletal and myocardial cells. The activation of mitochondrial CB1 receptor by its lipophilic agonists in these tissues was associated with mitochondrial regulation of the oxidative activity through relevant enzymes implicated in pyruvate metabolism [23]. Thereafter, melatonin receptors (MT1Rs) were recognized to localize in the outer mitochondrial membrane of neurons to regulate cell respiration. Strikingly, it was shown that the lipophilic melatonin ligand was produced in the mitochondrion to auto-regulate its MT1 receptors and in turn stimulate the Gαi proteins localized in the intermembrane space to inhibit stress-mediated cytochrome c release [45]. These remarkable findings challenge our classical perception of GPCRs’ biological function by showing that an intracellular organelle can both synthesize a ligand and directly respond to it through an auto-receptor mechanism. To name this amazing new mechanism, the term “automitocrine” was proposed, in analogy to “autocrine” when a similar phenomenon occurs among cells [45].

Notably, GPCRs are also present in cell division compartments where they regulate cytokinesis [24]. For instance, the odorant OR2A4 receptor localizes to the spindle poles during mitosis and to the cleavage furrow and midbody ring during cytokinesis in HeLa cells [24]. The crucial role played by these receptors in cell division was also established in OR2A4 knockdown experiments where the lack of the receptor caused cytokinesis failure.

Furthermore, over thirty GPCRs have been described to localize at the nuclear membrane and convincing evidence shows they play important physiological roles such as gene regulation [25,59,60,61]. In isolated nuclei from rat hearts, isoproterenol through the activation of β3 adrenergic receptors and Ang II through the activation of AT1 and AT2 receptors increase nuclear gene transcription. Importantly, these effects were blocked by the pertussis toxin (PTX), suggesting that the activation of Gi proteins was essential for β3 adrenergic and AT1 and AT2 receptor regulation of gene expression [62,63].

Unexpectedly, GPCRs have also been found in extracellular vesicles produced by eukaryotic cells, such as exosomes [51]. These tiny structures of 50–100 nm in diameter are formed in multivesicular intracellular bodies that are late endosomal compartments situated in the endocytic route between early endosomes and lysosomes. Internal vesicles of multivesicular bodies are generated by inward budding of the membrane and are released in the extracellular milieu following fusion of the multivesicular body with the plasma membranes. Exosomes can carry various types of proteins, lipids, and nucleic acids (mRNA and non-coding RNA) and have been recognized as important tools for cell-to-cell communication [64,65,66]. Specifically, Kwon et al. [51] found that GPRC5B, an orphan GPCR, is present in exosomes released by hepatocyte growth factor (HGF)-treated Madin–Darby canine kidney (MDCK) cell cysts. Exosomal GPRC5B is taken up by nearby MDCK cells and together with HGF promotes Erk phosphorylation and tubulogenesis. Furthermore, the same authors found that GPRC5B is elevated in urinary exosomes from patients with acute kidney injury, suggesting that the transport of this receptor through exosomes could recapitulate a repairing mechanism. After this initial finding, other GPCRs have been found to be secreted from cells via exosomes in various physiological and pathological contexts [67].

## 5. GPCR Sorting to the Plasma Membrane and to Intracellular Compartments

Distribution of proteins to different cellular compartments requires protein-sorting codes that are recognized and segregated by cytoplasmic adaptor complexes that regulate protein trafficking. Many proteins are sorted by short signal peptides attached to the N-terminus (or occasionally at the C-terminus or along the internal sequence) of the protein [68]. Integral membrane proteins including GPCRs are first synthesized in the perinuclear endoplasmic reticulum and then transported along the secretory pathway through the Golgi apparatus and the trans-Golgi network to be delivered to the plasma membrane. Once the nascent protein is inserted into the membrane, the signal peptide is normally cleaved off from the mature protein [69]. Most GPCRs lack a cleavable signal peptide and the molecular mechanisms that lead to their targeting to the plasma membrane, or their sorting to intracellular compartments, are poorly understood. Only a small group of GPCRs contains cleavable signal peptides and their removal results in the retention of the receptor in the endoplasmic reticulum [70]. The variety of the cellular destinations of GPCRs raises the question of how GPCRs are delivered to these targets.

Most of the work on GPCR trafficking has focused primarily on their plasma membrane localization and internalization. Several chaperone proteins bind to nascent GPCRs in the endoplasmic reticulum and carry them to the Golgi complex and finally to the plasma membrane [71]. Receptor activity-modifying proteins (RAMPs) are a family of three single pass membrane proteins that were initially discovered as regulators of the calcitonin receptor-like receptor (CLR) function and plasma membrane expression [72]. It is recognized that RAMPs also interact with several other GPCRs to switch ligand selectivity, and to modulate signal transduction and receptor trafficking [73,74]. Among others, the chaperone effects, first noted for CLR, have been shown for the calcium sensing receptor [75], the secretin receptor [76], the GPR30 receptor [77], and the type 1 corticotrophin releasing factor receptor (CRF1) [78].

Other proteins that have been shown to deliver GPCRs to the membrane are the receptor expression enhancing proteins (REEPs) and the receptor transporting proteins (RTPs), identified for their ability to enhance odorant and taste receptors’ cell surface expression [79]. Furthermore, the integral protein calnexin regulates the membrane expression of dopamine D1 and D3 receptors [80], CD4 enhances the plasma membrane expression of the chemokine CCR5 receptor [81], the transmembrane protein 147 (tmem147) reduces the M3 receptors at the membrane levels [82], and Rab43 regulates the expression of adrenergic α2B and muscarinic M3 receptors [83].

This brief overview indicates the heterogeneity in the molecular chaperones involved in GPCR trafficking and the lack of a common thread associated with this phenomenon. Furthermore, GPCR trafficking to the plasma membrane varies depending on the expression of the molecular chaperone and the context in which the two proteins (GPCR and chaperone) are expressed. In addition, the same GPCR can use different sorting mechanisms depending on the cell context in which it is expressed. For example, in primary neurons and in neuronal SH-SY5Y cells, Rab43 selectively regulates the total surface expression of the endogenous adrenergic α2 receptors, but not of the muscarinic M3 receptors. In contrast, the surface transport of both receptors requires Rab43 in non-neuronal NRK49F cells, suggesting that the sorting function of Rab43 is neuronal cell specific [83].

The nuclear membrane is another important target of GPCR sorting. In this compartment, GPCRs regulate nuclear events such as DNA synthesis and gene expression [63,84,85,86], transcription initiation [47], and histone modification [87]. The nuclear membrane is an extension of the endoplasmic reticulum and is formed by three connected membrane domains: the outer nuclear membrane that is a continuation of the ribosome-studded rough endoplasmic reticulum that also contains specific protein complexes; the pore membrane, where large macromolecular assemblies called nuclear pore complexes control the passage of molecules to and from the nucleus; and the inner nuclear membrane that faces the nucleoplasm and hosts a number of specific proteins that directly regulate the genome [88]. Like the plasma membrane, GPCR translocation in the inner nuclear membrane appears to be controlled by several processes including lateral diffusion through the membrane of the nuclear pore and those regulated by proteins of the canonical soluble protein transport machinery [41,88,89].

Some GPCRs localize to the nuclear membrane by using canonical nuclear localization signal (NLS) peptides, short basic sequences that confer specificity for one or more karyopherin nuclear transport proteins [90]. Karyopherin proteins were initially described as carriers of soluble proteins, but they are also recognized to play a major role in the transport of transmembrane proteins. The GPCRs that have been recognized to date to use this mechanism to localize to the nuclear membrane are the adenosine A1 and A2B, the angiotensin AT1, apelin, the bradykinin B2, CXCR2 and CXCR4, the coagulation factor II receptor-like 1 (F2rl1, previously known as Par2), and the oxytocin receptors [37,38,39,40,91,92]. Moreover, some GPCRs contain multiple NLS import sequences in different receptor parts. For example, F2rl1 has two NLS domains, in the first and third intracellular loops; mutations in either loop prevent nuclear translocation, suggesting that both are essential for karyopherin β1 binding [92]. Remarkably, this receptor has an additional C-terminal domain, that does not contain an NLS, but has a prominent role in nuclear transport. It probably interacts with proteins different from karyopherins that still concur with F2rl1 nuclear translocation. Other GPCRs instead translocate to the nucleus though a phosphorylation-mediated mechanism as shown for the glutamate mGlu5 receptor [93].

Interestingly, the Ras superfamily proteins of small GTPases, such as Rab and Arf GTPases, have also emerged as crucial regulators of GPCR localizations [94]. In particular, they control vesicular trafficking, vesicular budding from donor membranes, interactions with cellular motors, and vesicle docking. They are networked to one another through a variety of mechanisms to coordinate the individual events of one stage of transport and to link together the different stages of an entire transport pathway [95]. Among others, Rab11a plays a pivotal role in agonist-independent nuclear translocation of the platelet-activating factor receptor [96]. Interestingly, emerging evidence suggests that several family members of the Ras and Rho small GTPases have putative NLSs. The most prudent assumption is that these proteins complex to GPCRs on one side and to canonical nuclear transport proteins on the other to translocate GPCRs to the nucleus [97].

In general, GPCRs that localize on endosomes, endoplasmic reticulum, and Golgi and outer nuclear membranes have their N-terminus embedded in the lumen of these structures with the carboxyl terminal facing the cytoplasm (Figure 1). On the contrary, activated GPCRs in the inner nuclear membrane signal into the nucleoplasm and directly influence nuclear functions [88]. In mitochondria, GPCRs have been localized in the outer [44,45,98,99] and inner membranes [43] and they seem to be oriented with their signaling part toward the intermembrane space and the matrix, respectively.

GPCR trafficking to internal compartments can be independent from endocytosis, as it is for the platelet-activating factor receptor (F2RL) that can directly reach the nuclear localization through the trans-Golgi network [96] or receptor internalization in endosomes. While the former is normally agonist independent, the latter is an agonist-dependent process [94].

## 6. β-Arrestin-Mediated GPCR Compartmentalization

The binding of agonists to GPCRs is characterized by key conformational changes necessary for G protein-dependent signaling transduction and for the exposure of phosphorylation sites to kinases such as GRKs. GPCR phosphorylation is a crucial step for G protein-dependent signal desensitization, leading to the uncoupling of G proteins from the receptor and the recruitment of the β-arrestin proteins. β-Arrestins are crucial for receptor removal from the plasma membrane and thus for the regulation of GPCR endocytic trafficking.

The “barcode” model of β-arrestin/GPCR interaction suggests that the degree and patterns of GPCR phosphorylation match different β-arrestin structural changes, and this matching favors different intracellular signaling [100]. The pattern of phosphorylation directly affects the interaction with the β-arrestin family as described for the β2 adrenergic receptor phosphorylated by GRKs [101,102]. Moreover, besides the canonical GRKs, GPCR phosphorylation could also be mediated by more versatile kinases such as casein kinase 2 (CK2), a kinase that plays per se a crucial role in the cell cycle. In particular, CK2 regulates the M3 muscarinic receptor activity by direct phosphorylation and in β-cells it affects the M3 receptor’s ability to favor insulin secretion [103,104,105]. Protein kinase C (PKC) is another GPCR-phosphorylating enzyme that regulates β-arrestin recruitment as shown for the chemokine receptor CXCR4, at Ser-346 and 347 after agonist stimulation [106]. Therefore, GPCR phosphorylation and the ability to recruit β-arrestin to signal are highly intertwined, especially for their ability to evoke specific intracellular signaling such as Erk phosphorylation, desensitization, and antiapoptosis effects [101].

In cancer and healthy cells, these GPCR–β-arrestin-dependent multiprotein complexes interact with signaling proteins involved in gene transcription, protein ubiquitination, and cytoskeletal remodeling, forming signalosomes. These large supramolecular complexes promote cancer progression and metastasis production by activation of mitogen-activated protein kinase/extracellular signal-regulated kinase, Wnt/β-catenin, nuclear factor κB, and phosphoinositide 3 kinase/Akt [107]. Several in vitro systems have recently been developed to investigate radio- and chemotherapy-resistant cancer cells. In particular, cancer cell lines were exposed to drugs or radiation with the aim of selecting treatment-resistant clones and thus analyze the processes of cancer therapy resistance [108,109]. Notably, based on the major roles played by GPCR–β-arrestin signalosomes in regulating cell growth and survival, the mentioned in vitro approaches could lead to the identification of the GPCR–β-arrestin complex-based mechanisms that promote cancer chemo- or radiotherapy resistance. Intriguingly, it has been observed that the overexpression of GPR35 receptor strongly correlates to drug resistance in epithelial lung cancer cells [110].

While in cancer the GPCR–β-arrestin signalosomes play a crucial role in promoting disease progression, in neurodegeneration they are beneficial by slowing down the development of misfolded proteins involved in neurodegenerative disorders. This is the case of the M1 muscarinic receptor and its ability to efficiently signal through β-arrestin [111,112]. Mutant mice with phosphorylation-deficient M1 receptors have more rapid and pronounced misfolded prion-mediated neurodegeneration progression than controls. This strongly suggests that the M1–β-arrestin complex signal has important neuroprotective effects (Figure 2). Therefore, the next generation of GPCR ligands designed to directly modulate GPCR–β-arrestin-dependent intracellular signaling could pave the way towards, for example, the development of novel neuroprotective and anticancer strategies.

## 7. Activation of GPCRs in Internal Cell Compartments

One of the critical issues when considering GPCRs localized in internal compartments is how they are activated by endogenous ligands (Figure 3).

Apart from small solutes of moderate polarity, the number of natural molecules that can passively diffuse across the plasma membrane is surprisingly limited, and among them we can recognize the steroid hormones [113] and melatonin [114]. For instance, extracellular melatonin equilibrates with the cytoplasm with a half time of about 3.5 s while the different steroid hormones equilibrate with half times ranging from 10 to 20 s [115]. On the other hand, charged small natural ligands such as noradrenaline and 5-hydroxytriptamine need 5 and 1.5 h, respectively, to equilibrate, suggesting that they reach the internal cell compartments by other means than passive diffusions [114]. Bioactive lipids such as endocannabinoids, prostaglandins, and sphingolipids, just to name a few, in principle could cross the plasma membrane due to their lipophilicity, but for each of these ligands one or more specific transporters have been identified. The transporters take up these signaling molecules and in combination with other proteins and carriers deliver them to specific intracellular organelles for signaling and/or degradation [116,117,118]. Anandamide, for instance, crosses the plasma membrane though the endocannabinoid membrane transporter (EMT) and following internalization it interacts with cytosolic carriers (albumin, HSP70, FABP5) to be degraded by hydrolase or lipase enzymes or to reach CB1 receptors on the mitochondria (Figure 3).

In fact, it has been shown that prevention of bioactive lipid degradation by the dual inhibition of fatty acid amide hydrolase and monoacylglycerol lipase (EDE), the two main endocannabinoid-degrading enzymes, boost the neuronal mitochondrial cannabinoid CB_1_ receptor signaling that contributes to the endocannabinoid-dependent depolarization-induced suppression of inhibition in the hippocampus [119].

Other natural ligands that can activate their intracellular GPCR targets following uptake are small molecules such as glutamate and catecholamine. Glutamate can activate nuclear mGlu1 receptors in rat cortical nuclei after uptake by the sodium-dependent excitatory amino acid transporters and the cystine/glutamate exchanger. Importantly, the inhibition of these transporters can interfere with the intracellular receptor activation [120,121].

With respect to catecholamines, it is unlikely that the classic reuptake system plays a role in carrying these small molecules to intracellular receptor targets, as it has mainly a role in terminating the synaptic signaling at neuronal terminals. However, non-selective uptake systems such as the extraneuronal monoamine transporter could transport these molecules across the plasma membrane [122,123]. For example, following incubation of neonatal rat ventricular myocytes with [3H]noradrenaline, Buu et al. [124] observed a time-dependent intracellular accumulation of this amine with the highest proportion recovered in the nuclear fraction. Previously, these authors demonstrated the presence of α1 and β1 adrenergic receptors in isolated cardiomyocyte nuclei by binding assay and suggested that intracellular noradrenaline could bind these targets [124]. It has also been shown that OCT3, the major component of the extraneuronal monoamine transport system, takes noradrenaline inside the cells to activate intracellular adrenergic receptors such as the nuclear α1 adrenergic receptors in adult cardiac myocytes or the β1 adrenergic receptors at the inner nuclear membrane in astrocytes [125,126,127].

In analogy to the autocrine activation of a receptor on the plasma membrane, we could generally talk of “autointracrine” activation when a cell produces a natural ligand that acts on a receptor inside the cells. We have already mentioned this effect for melatonin MT1 receptors located in mitochondria (Figure 3) [45], but it has been shown that other natural ligands such as prostaglandins and platelet-activating factor of lysophosphatidic acid show these types of autointracrine regulation system [128]. For instance, biogenesis of prostaglandins depends on the sequential action of cyclooxygenases and prostaglandin synthetase enzymes on the arachidonic acid released by phospholipase A2 from the plasma membrane. All these enzymes can be found on the nuclear envelope to locally direct the formation of prostaglandins and regulate the activity of nuclear, localized prostanoid receptors [129,130].

For natural peptide ligands, crossing the plasma membrane and targeting intracellular GPCRs is difficult in terms of size and charges; however, it has been shown that small peptides, such as the melanostatin or thyrotropin-releasing hormone (TRH), could in principle use the peptide transporters to cross the cellular membrane and reach intracellularly compartmentalized receptors [131]. These transporters are integral membrane proteins that uptake di- and tri-peptides: PEPT1 is the low-affinity, high-capacity transporter and is mainly expressed in the small intestine, PEPT2 is the high-affinity, low-capacity transporter and has a broader distribution in the body. Expression of PEPT2 has been shown in glia [132] and evidence for the uptake of TRH by glial cells has been provided by Pacheco et al. [133]. Nevertheless, this mechanism seems to mainly have a role of clearance of these di- and tri-neuropeptides. However, chances are that in some cases the uptake of active short chain peptides could bypass degradation and stimulate intracellular GPCRs.

In any case, it is more likely that natural peptide ligands can reach the intracellular compartment by internalization with their cognate receptors. As a rule, in early endosomes, after internalization, hormone peptides dissociate by their cognate receptors to be degraded in lysosomes, while the receptors are recycled to the cell membrane [134,135]. Nevertheless, the ligand/receptor complex could continue its journey to an intracellular compartment. This is the case of oxytocin that in osteoblasts, breast cancer cells, and primary fibroblasts internalizes with its receptor and reaches the nuclear membrane (Figure 3) [40,136].

Constitutive activity is another way GPCRs could activate signaling in intracellular compartments. Many GPCRs exhibit constitutive activity [137,138] and the different compositions of the membrane in the different cellular compartments could affect it. As an example, cholesterol can positively or negatively affect the activity of many GPCRs [139] and its amount is clearly higher in the plasma membrane, where it reaches 60% to 80% of total cellular cholesterol, compared to other cellular membranes [140]. Another mechanism of ligand-independent stimulation of GPCRs is by association with proteins that switch on GPCR activity by direct interaction, as has been shown for Homer1a protein that leads to the agonist-independent activation of mGlu5 receptors [141].

## 8. Distinctive Signals Generated by GPCRs in Intracellular Compartments

The biggest limitation to explore GPCR intracellular signaling is represented by the fact that cells need to be lysed and this procedure clearly directly affects the temporal and spatial resolution of the biochemical events of these phenomena. A major breakthrough in the field was achieved with the development of fluorescent and bioluminescent resonance energy transfer (FRET/BRET) sensors [142]. These RET sensors allow monitoring of GPCR signaling in real time without altering the cell homeostasis.

The first RET sensor to be developed was for the second messenger adenosine 3′,5′-cyclic monophosphate (cAMP). The sensor consisted of cAMP-dependent protein kinase in which the catalytic and regulatory subunits were each labeled with a different fluorescent dyes able to generate FRET signals when complexed to the holoenzyme. Binding of cAMP to the holoenzyme induced subunit dissociation and the elimination of the FRET signal. The change in shape of the fluorescence emission spectrum allowed the quantification of cAMP concentrations in single living cells [143]. In 2009, by using an Epac cAMP FRET biosensor, Calebiro et al. [49] demonstrated that the thyroid-stimulating hormone (TSH) receptor can generate sustained cAMP signal after internalization and Ferrandon et al. [32] showed the same effect for the internalized parathyroid hormone (PTH) receptors. Interestingly, the effect of the TSH receptor was cell specific as it occurred in primary thyroid cells but not in HEK293 cells [144], while the stimulation of the internalized PTH receptor was agonist specific as it was stimulated by PTH1-34, but not by PTH-related peptide (PTHrP1-36) [32]. The substantial difference between PTH1-34 and PTHrP1-36 is that the former displays an exceptional ability to stabilize an active state of the receptor and to also remain associated with it for a long time when the receptor is internalized in endosomes, while the latter rapidly dissociates, especially after internalization [145]. Another example of an intracellular GPCR that is differently stimulated by agonists with distinct binding strength is the vasopressin V2 receptor. The arginine vasopressin agonist, that binds tightly to the V2 receptor, induces a prolonged G protein signaling in endosomes, whereas oxytocin that binds with lower affinity results in predominantly plasma membrane receptor activation signaling [34]. These characteristics, evidently, can be exploited to construct drugs targeting specifically intracellular GPCRs.

Biosensors generated to measure inositol-1,4,5-trisphosphate (InsP3) [146] and diacylglycerol [147] have demonstrated the production of these second messengers by the activation of intracellular GPCRs, as well. Furthermore, FRET-based calcium biosensors, with a reduced number of calcium binding sites per sensor, have been optimized to allow visualization of tonic action potential firing in neurons and high-resolution functional tracking of T lymphocytes [148].

Other biosensor types can directly measure GPCR activation [149]. For example, Irannejad et al. [26] examined the subcellular distribution of the activated β2 adrenergic receptor using the nanobody-based translocation sensor Nb80-GFP. This sensor selectively recognizes the β2 adrenergic receptor active forms. Strikingly, after a prolonged stimulation with the agonist isoprenaline, Nb80-GFP signal was localized to the intracellular puncta that contained internalized, endosomal β2 adrenergic receptors. Furthermore, these types of sensors have shown that β1 adrenergic receptors localized in the Golgi membrane [150].

Persistent activation of GPCRs in endosomes after internalization contrasts with the common notion that the interactions of GPCRs with β-arrestins and G proteins are mutually exclusive. This intriguing issue was solved by Thomsen et al. [151] who demonstrated that a single GPCR can simultaneously bind through its core region with G protein and through its phosphorylated C-terminal tail with β-arrestin, forming a super-complex or “megaplex”. These results established a novel paradigm in the GPCR signaling in which the interaction of G protein and β-arrestin with the activated receptor is no longer limited to events at the plasma membrane levels but also in intracellular compartments. Recent studies uncovered an even more complex scenario of GPCR signaling in endosomes, with the β2 adrenergic receptor promoting endosomal cAMP production mediated by the PTH receptor through the stimulatory action of Gβγ protein subunits on adenylate cyclase type 2, which in turn resulted in a prolonged nuclear PKA activation and CREB phosphorylation [152]. These studies raise the intriguing possibility that the cAMP generated in endosomes has a distinct role compared to cAMP generated at the plasma membrane, for instance, to activate the PKA within specific intracellular compartments, such as the nucleus [153]. These results reveal that the complexity of GPCR signaling is much greater than currently appreciated. The rapidly evolving technology of RET sensors will allow us to gain deeper insights into the emerging field of intracellular GPCR signaling and their physio- and patho-physiological roles.

## 9. Designing Drugs to Target GPCRs Localized in Internal Compartments

Receptors in internal compartments may represent a potential drug target with possible therapeutic relevance. Lipophilic ligands might have access to intracellularly compartmentalized GPCRs by simple diffusion throughout the plasma membrane, whereas hydrophilic compounds could be transported across the cell membranes by specific proteins (Figure 3 and Figure 4). In 1997, our group showed how the highly lipophilic dopamine agonist pergolide can reach and activate intracellular dopamine D2 receptors, suggesting that ligands could have distinctive pharmacological profiles based on their ability to exclusively activates GPCRs on the cell surface or intracellularly [154]. Strikingly, twenty years later Shioda et al. [155] reported a novel signaling pathway through the intracellularly localized long isoform of the dopamine D2 receptor (D2L). In particular, they showed that when localized in the Golgi, D2L elicits Gαi3-mediated Erk signaling and dendritic spine formation through Rabex-5/platelet-derived growth factor receptor-β (PDGFRβ). Furthermore, they showed that dendritic spine density in striatopallidal medium spiny neurons significantly increased following treatments of striatal slices with the lipophilic agonist quinpirole; importantly, those changes were lacking in D2L knockout mice. This evidence clearly suggests a role for dopamine receptors that go beyond the extrapyramidal motor regulation and the induction of psychotic symptoms such as delusions and hallucinations. This could in fact be relevant in neuropsychiatric disorders such as schizophrenia that has been conceptualized as a disorder of connectivity and abnormal synaptic modeling [156,157]. In this respect, antagonists that reach dopamine receptors inside the cells could re-establish a normal connectivity among neurons compared to drugs with less propensity to reach intracellular receptors. As a matter of fact, the most effective atypical antipsychotic, clozapine [158,159], has a higher lipid membrane penetration coefficient than the typical antipsychotics chlorpromazine and haloperidol [160], indicating that clozapine’s peculiar pharmacological profile could depend in part on the effective antagonist concentrations reaching inside the neurons. Similar findings were also reported for olanzapine, another highly effective atypical antipsychotic, when compared with chlorpromazine [161], suggesting that one of the differences between typical and atypical antipsychotics could be correlated to the efficacy of the latter to antagonize intracellular dopamine receptors.

Another way these drugs could cross the plasma membrane is by carrier-mediated intracellular transport. In humans, more than 300 solute carrier (SLC) transporters have been identified and these mainly include organic anion-transporting polypeptides (OATPs/SLCOs, organic anion transporters (OATs/SLC22As), organic cation transporter (OCTs/SLC22As), organic cation and carnitine transporters (OCTNs/SLC22As), peptide transporters (PEPTs/SLC15As), and multidrug and toxin extrusions (MATEs/SLC47As) (for a broad review of the argument, see [162] and articles in the same volume). Most SLC transporters belong to influx transporters and mediate movement of drugs from the extracellular to the intracellular compartments, either by passive diffusion along the drug concentration gradient, by cotransport, or counter-transport against its concentration gradient and by co-opting the concentration gradient of another solute. SLC transporters are known to transport countless drugs and they could be exploited to target intracellular GPCRs. On the other end, drug efflux mediated by the ABC transporter family could have a counter-effect by reducing intracellular drug concentrations [163]. In fact, these proteins are best known for their contributions to chemoresistance through the efflux of anticancer drugs. In this context, inhibitors of these efflux transporters could increase the concentration of drugs in intracellular compartments [164].

Finally, drugs could target intracellular GPCRs by conjugation with compounds that increase their rapid incorporation into the plasma membrane, such as polyethylene glycol or the transmembrane lipid cholestanol [165]. For example, antagonists of neurokinin type 1 (NK1), spantide, and the calcitonin receptor-like receptor (CLR) CGRP8–37, when conjugated with polyethylene glycol and cholestanol, inhibit signals that originated from endosomal NK1 and CLR receptors emphasizing the physiological importance of GPCR signaling from endosomes (Figure 4) [27,31]. These conjugated compounds, intrathecally injected, have also been shown to inhibit mechanical nociceptive responses, suggesting their development as pain relivers. Drugs could also be designed to reduce/inhibit the recycling of GPCRs to the plasma membrane, thus increasing their intracellular pool. The atypical antipsychotic clozapine, unlike other antipsychotic drugs such as haloperidol that increase the number of D2 receptors on the plasma membrane, reduces their translocation from the intracellular pool to the cell surface [166]. This effect of clozapine does not seem to be correlated with its fast dissociation kinetics, low affinity, and transient occupancy of the D2 receptor [167], but it seems to be correlated to a negative pharmacoperone effect. All these considerations support the possibility to develop drugs targeting GPCRs localized in intracellular compartments.

## 10. Concluding Remarks and Future Outlook

In the last decade, the plasma membrane-centric view of GPCR signaling was confuted with accumulating data showing that intracellular GPCRs can signal in the same way as those in the plasma membrane. Intracellular GPCRs activate unique pathways, different from those activated on the plasma membrane, and/or they prolong biological responses starting at the plasma membrane [86]. Furthermore, intracellular GPCRs can intervene in many physiological and pathological processes. Activation of GPCRs within subcellular compartments is not a bizarre phenomenon of minor significance but is an essential cellular mechanism to confine signals and separate metabolic processes that would otherwise interfere with each other. For instance, the transport of the receptor to internal compartments, such as the nucleus or the mitochondria, the activation of internally localized GPCRs that requires endogenous molecules to cross the plasma membrane and reach the receptors inside, and the convergence of GPCR effectors, such as G proteins, in the same intracellular compartment, are all well-orchestrated mechanisms that have required a finely coordinated evolutionary process. We are just at the beginning in understanding this complex GPCR signaling and many questions still need to be answered, such as: is there any way to selectively activate or inhibit intracellular GPCRs without interfering with cell surface receptors? How long would a drug targeting intracellular GPCR remain inside the cell and stimulate or block the receptor, given the slower clearance? Are there mechanisms of down- or up-regulation like the GPCRs on the cell surface?

However, the evidence reported in this review is already compelling regarding the possibility that compartmentalized GPCR signaling will soon be an important pharmacological target for the development of novel and more effective drugs.

## Figures and Tables

**Figure 1 biomolecules-12-01343-f001:**
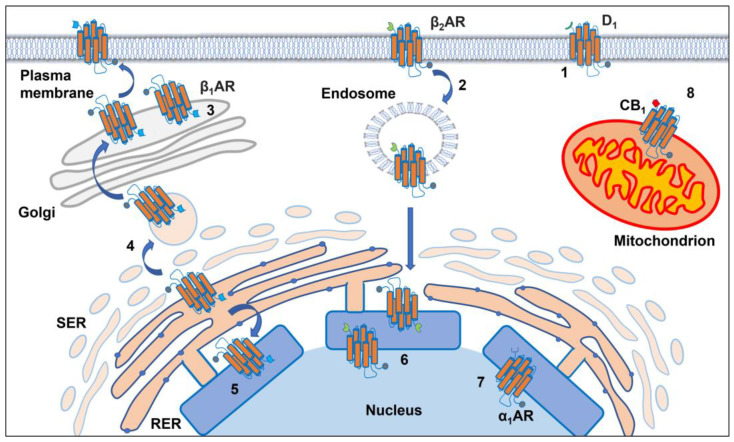
GPCRs’ main cell localization. GPCRs are represented as orange proteins with carboxyl groups shown as gray dots and N-terminal groups with symbols of different colors, blue, green, red. Canonically functional GPCRs localize at the plasma membrane level (“1”), but they can also be functional on the membrane of different intracellular compartments: endosomes where their carboxyl domains face the cytosol (“2”); the Golgi apparatus, where they constitute a pre-existing receptor pool and to where they are delivered after being assembled in the perinuclear endoplasmic reticulum (“3”, “4”, respectively); the nucleolus, on the outer and inner nuclear membrane where the GPCR carboxyl domains face the cytosol or the nucleoplasm, respectively (“5”, “6”). GPCRs can also form pre-existing pools in the nuclear inner membrane with their carboxyl domain facing the nucleoplasm (“7”). Finally, GPCRs can be found in the mitochondrial outer membrane where their GPCR carboxyl domains face the intermembrane space (“8”).

**Figure 2 biomolecules-12-01343-f002:**
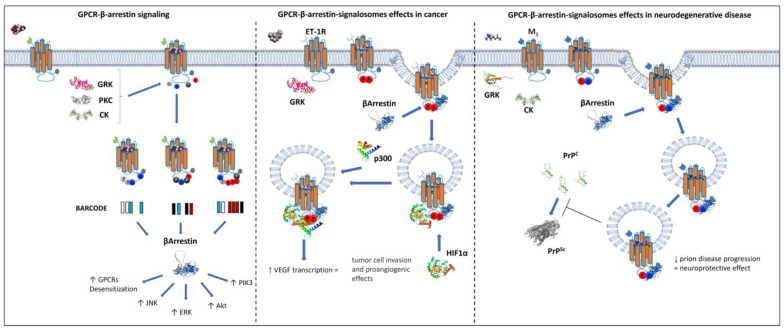
GPCR/β-arrestin-dependent signaling pathway. GPCRs are illustrated as orange transmembrane proteins whose carboxyl groups are shown as gray dots and their N-terminal groups as differently shaped symbols. GPCR phosphorylation processes can take place through the action of GPCR kinases (GRKs) or casein kinase (CKs), represented with red and blue dots, respectively. β-arrestin functions are to reduce GPCR coupling to G proteins, favor the internalization of the receptor in endosomes and signal through the GPCR-β-arrestin complex to promote JNK, ERK, phosphoinositide 3 kinase (PIK3), and Akt signaling. Different degrees of GPCR phosphorylation affect the interaction between receptor and β-arrestin, thus initiating preferential intracellular responses such as β-arrestin mediating ERK, JNK, or GPCR desensitization (barcode theory). In cancer cells, these GPCR–β-arrestin-dependent multiprotein complexes interact with signaling proteins involved in gene transcription: in ovarian cancer, the activation of ET-1R promotes the interaction between β-arr1/p300 and HIF-1α, enhancing the transcription of genes, such as ET-1 and VEGF, required for tumor cell invasion and proangiogenic effects. Meanwhile, in neurodegenerative disease, GPCR–β-arrestin signalosomes exert a crucial neuroprotective effect as shown in the right part of the figure, where the complexed, internalized muscarinic M_1_ receptor signaling reduces the accumulation of misfolded prion protein (PrPsc).

**Figure 3 biomolecules-12-01343-f003:**
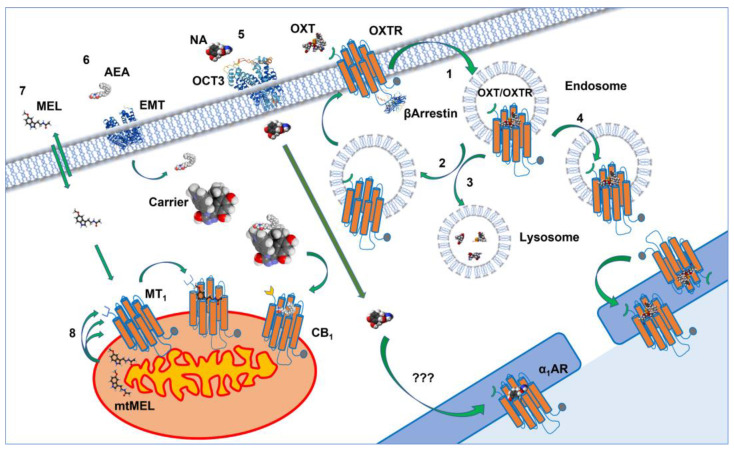
Extra- and intracellular GPCR activation by endogenous ligands. GPCRs are represented as orange proteins whose carboxyl groups are shown as gray dots whereas their N-terminal groups are differently shaped symbols of different colors. Canonically, ligands activate GPCRs at the plasma levels (“1”) where they promote internalization as shown for oxytocin receptors (OXTR). Internalized receptors are recycled (“2”) and their ligand degraded in lysosomes (“3”) or translocated in complex with their ligand to the outer and inner nuclear membrane (“4”). Some GPCRs embedded in the inner nuclear membrane are activated by hydrophilic ligands that cross the plasma membrane through transporters (organic cation transporter 3, OCT3) such as noradrenaline (NA) that activates the nuclear compartmentalized adrenergic receptors (“5”). Additionally, highly lipophilic ligands such as anandamide (AEA) cross the plasma membrane through transporters (EMT) and reach the cannabinoid receptors on the outer mitochondrial membrane through intracellular carriers, regulating cell respiration (“6”). Some ligands such as melatonin reach their intracellular GPCR targets by crossing the plasma membrane through simple diffusion and by an “automitocrine” mechanism in which the intracellular organelle itself synthesizes the ligand, autoregulating its own functions (“7”, “8”, respectively).

**Figure 4 biomolecules-12-01343-f004:**
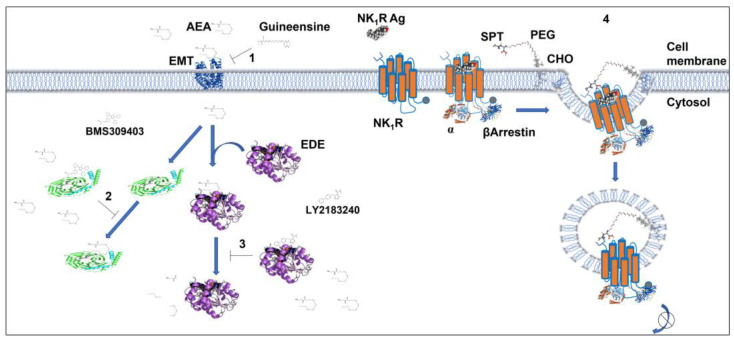
Compartmentalized GPCRs as novel pharmacological targets. The endogenous ligand anandamide (AEA) after uptake by the endocannabinoid membrane transporter (EMT) interacts with cytosolic proteins such as albumin, HSP70, FABP5 and is degraded by fatty acid amide hydrolase and monoacylglycerol lipase enzymes (EDE). These steps can be pharmacologically targeted to decrease (guineensine, “1”; BMS309403, “2”) or increase (LY2183240, “3”) the intracellular levels of AEA, modulating its compartmentalized intracellular signaling effects. Intracellular GPCR signaling is also blocked by conjugation of receptor-specific antagonists. NK1R is depicted as a seven-transmembrane orange protein, whose gray dots represent carboxyl groups, and the N-terminal group is a blue symbol. Antagonists of neurokinin type 1 (NK1) and spantide (SPT), when conjugated with polyethylene glycol (PEG) and cholestanol (CHO), inhibit signals that originate from endosomal NK1 receptors emphasizing the physiological importance of GPCR signaling from endosomes (“4”).

**Table 1 biomolecules-12-01343-t001:** List of GPCRs that signal from intracellular compartments.

Subcellular Localization	Receptor	Reference
Early endosomes	β2-adrenergic receptor (β2AR)	[26]
	Calcitonin-gene-related-peptide receptor (CGRPR)	[27]
	Calcium-sensing receptor (CaSR)	[28]
	Dopamine receptor type 1 (D1R)	[29]
	Luteinizing hormone receptor (LHR)	[30]
	Neurokinin type 1 receptor (NK1R)	[31]
	Parathyroid hormone receptor (PTHR)	[32]
	Protease activated receptor 2 (PAR2)	[33]
	Vasopressin type 2 receptor (V2R)	[34]
Nucleus	α1A-adrenergic receptor (α1A-AR)	[35]
	α1B-adrenergic receptor (α1B-AR)	[35]
	Adenosine A1 receptor (ADORA1)	[36]
	Adenosine A2B (ADORA2B)	[36]
	Angiotensin AT1A receptor (AT1AR)	[37]
	Apelin receptor (APJ)	[38]
	Bradykinin B2 receptor (BKR2)	[38]
	Cysteine (C)-x-C receptor 4 (CXCR4)	[39]
	Oxytocin receptors (OXTR)	[40]
	C-C chemokine receptor type 2 (CCR2)	[41]
	Arginine vasopressin receptor 1α (AVPR1a)	[41]
	Sphingosine 1-phosphate receptor 1(S1P1)	[41]
Mitochondria	Purinoceptor 1 like receptor (P2Y1)	[42]
	Purinoceptor 2 like receptor (P2Y2)	[42]
	Angiotensin II receptor type 1 (AT1R)	[43]
	Angiotensin II receptor type 2 (AT2R)	[43]
	5-hydroxytrptamine receptor (5-HTR3 and 5-HTR4)	[44]
	Melatonin MT1 receptor (MT1R)	[45]
	Cannabinoid type 1 receptor 1 (CB1R)	[46]
Golgi	β1-adrenergic receptor (β1AR)	[47]
	Sphingosine-1-phosphate 1 receptor (S1P1R)	[48]
	Thyroid stimulating hormone receptor (TSHR)	[49]
ER	G Protein-Coupled Estrogen Receptor 1 (GPR30)	[50]
	Metabotropic glutamate receptor 5 (mGluR5)	[25]
Exosomes	G Protein-Coupled Receptor Class C Group 5 Member B (GPRC5B)	[51]
Spindle poles	Olfactory receptor 2A4 (OR2A4)	[24]

## Data Availability

Not applicable.

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
