# Peer review of "GPCRs in Intracellular Compartments: New Targets for Drug Discovery"

_biomolecules, 2022, doi:10.3390/biom12101343_

Round 1

Reviewer 1 Report

In this manuscript the authors review some of the current concepts regarding subcellular localisation of GPCRs and their consequences for signalling.

Overall, this is a well written review that highlights the existence of receptor locations distinct than the plasma membrane.

I have two general comments about this manuscript. Firstly, while the authors make the effort to present a significant amount of literature, they should emphasise in different locations within the text that they are illustrating concepts with examples, to avoid the view that the receptors cited are the only cases in the literature and that indeed there are more receptors subject to the regulations presented. Secondly, while the authors illustrate the existence of different subcellular localisations for GPCRs, they do not emphasise whether this is relevant for physiology, and the importance of subcellular signalling remains somewhat diluted in the content. Perhaps for each case presented here it would be of benefit to expand on the physiological importance and the therapeutic relevance.

Specific comments:

·      Abstract needs further proofreading (e.g. line 17 should be ‘viewed’; line 19 should be ‘were’, and line 21 should be ‘localize’).  

·      Lines 109-112: this statement on the location of the binding site and the coupling of effectors should be revised. Many GPCRs have the ligand binding site within the TMD. I suggest rephrasing focusing on the intracellular site of the TMD and intracellular loops.

·      Line 162 Figure 1 legend – should be “being”

·      Line 258 and below should be “chaperone”

·      Paragraph starting line 280 should be rephrased, as it is, it is confusing.

·      Line 348 – it is the identification of degrees and patterns of phosphorylation that brought up the concept of the barcode. Similarly, in the same paragraph, mentions to other kinases (e.g PKC) are omitted.

·      Figure 2 legend should be expanded. The figure does not completely depict GPCR-b-arrestin dependent signalign pathways, but two selected and specific examples.

·      Figure legends for Figure 3 and 4 seem particularly long and repetitive from the text. These need to be revised.

·      Line 514 – It is unclear why constitutive activity of GPCRs is brought up in this part of the manuscript.

·      Line 524 – this section does not acknowledge the existence of calcium sensors a long time before the cAMP sensors were generated

·      Page 615 – recent descriptions of clozapine as chaperone for D2 are not mentioned (PMID: 30670597).

Reviewer 2 Report

Overall this is a well-written and well-organized review by Fasciani et al. I have some comments and questions for the authors.

- Section 9. While drug entry could be promoted by targeting SLC transporters, have the authors considered how the presence of efflux pumps could have a counter-effect?

- Have the authors considered how the pharmacological chaperone effect of small molecule GPCR ligands might add complexity to therapeutic modulation of GPCRs (surface and intracellular)?

- Could the authors elaborate on newer techniques for intracellular drug delivery that could be potentially applied to deliver drugs to intracellular GPCRs?

Minor issues

Line 57: Should be “evolutionary”

Line 58: Should be “complex organisms”

Line 68: Environmental

Line 81: raises

Line 95: functionally separate

Line 111: “while the transmembrane core shafts to the coupling to intracellular transducers”. Please edit this phrase.

Line 124: Omit “the”

Line 135: substantial number of targets, such as enzymes and channels

Line 188: have been known

Line 237: authors, not Authors

Line 247: synthesized

Line 258: chaperone

Line 265: calcium

Line 276: chaperones

Line 309: mutations in either loop

Line 316: Ras

Line 325: parsimonious is an odd word choice. Please edit the sentence for clarity.

Line 336-337: Rephrase for clarity

Line 340: β-Arrestin

Line 365: in vitro should be in italics

Line 406: half life or half time?

Line 418: Should it be albumin?

Line 459: burst is an odd word choice. Please edit the sentence.

Line 475: authors

Line 519: remove the word than

Line 529-530: rephrase sentence for clarity.

Line 578: raise

Line 612: highly effective

Line 627: remove the word a
